# Peripheral Immune Profiles in Individuals at Genetic Risk of Amyotrophic Lateral Sclerosis and Alzheimer’s Disease

**DOI:** 10.3390/cells14040250

**Published:** 2025-02-10

**Authors:** Laura Deecke, Olena Ohlei, David Goldeck, Jan Homann, Sarah Toepfer, Ilja Demuth, Lars Bertram, Graham Pawelec, Christina M. Lill

**Affiliations:** 1Institute of Epidemiology and Social Medicine, University of Münster, Albert-Schweitzer-Campus 1, 48149 Münster, Germany; laura.deecke@uni-muenster.de (L.D.); jan.homann@uni-muenster.de (J.H.); 2Department of Immunology, University of Tübingen, 72076 Tübingen, Germanygraham.pawelec@uni-tuebingen.de (G.P.); 3Department of Endocrinology and Metabolic Diseases (Including Division of Lipid Metabolism), Charité–Universitätsmedizin Berlin, Corporate Member of Freie Universität Berlin and Humboldt-Universität zu Berlin, Augustenburger Platz 1, 13353 Berlin, Germany; 4BCRT—Berlin Institute of Health Center for Regenerative Therapies, Berlin Institute of Health, Charité—Universitätsmedizin Berlin, 10117 Berlin, Germany; 5Lübeck Interdisciplinary Platform for Genome Analytics (LIGA), University of Lübeck, 23562 Lübeck, Germany; 6Health Sciences North Research Institute of Canada, Sudbury, ON P3E 2H3, Canada; 7Ageing and Epidemiology Unit (AGE), School of Public Health, Imperial College London, London W6 8RP, UK

**Keywords:** immune system, immune cell, neurodegeneration, polygenic risk score, disease prediction

## Abstract

The immune system plays a crucial role in the pathogenesis of neurodegenerative diseases. Here, we explored whether blood immune cell profiles are already altered in healthy individuals with a genetic predisposition to amyotrophic lateral sclerosis (ALS) or Alzheimer’s disease (AD). Using multicolor flow cytometry, we analyzed 92 immune cell phenotypes in the blood of 448 healthy participants from the Berlin Aging Study II. We calculated polygenic risk scores (PGSs) using genome-wide significant SNPs from recent large genome-wide association studies on ALS and AD. Linear regression analyses were then performed of the immune cell types on the PGSs in both the overall sample and a subgroup of older participants (>60 years). While we did not find any significant associations between immune cell subtypes and ALS and AD PGSs when controlling for the false discovery rate (FDR = 0.05), we observed several nominally significant results (*p* < 0.05) with consistent effect directions across strata. The strongest association was observed with CD57+ CD8+ early-memory T cells and ALS risk (*p* = 0.006). Other immune cell subtypes associated with ALS risk included PD-1+ CD8+ and CD57+ CD4+ early-memory T cells, non-classical monocytes, and myeloid dendritic cells. For AD, naïve CD57+ CD8+ T cells and mature NKG2A+ natural killer cells showed nominally significant associations. We did not observe major immune cell changes in individuals at high genetic risk of ALS or AD, suggesting they may arise later in disease progression. Additional studies are required to validate our nominally significant findings.

## 1. Introduction

The development of neurodegenerative diseases (NDs) arises from a complex interplay between genetic and environmental factors. Amyotrophic lateral sclerosis (ALS) is characterized by the degeneration of both first and second motor neurons, resulting in severe symptoms of muscle weakness and ultimately neuromuscular respiratory failure, making it one of the most fatal forms of NDs [1]. ALS occurs 1.5 times more frequently in men than in women, and men tend to be diagnosed about five years earlier than women [2]. Alzheimer’s Disease (AD), the most common ND, is characterized by the accumulation of amyloid-beta and tau proteins in the brain, ultimately leading to cognitive decline and dementia [3]. Two-thirds of AD patients are women, and their lifetime risk of developing the disease is higher (1 in 5) compared to men (1 in 10) [4]. It has been suggested that this is not only explained by women’s longer lifespan, but also by the biological impact of sex on the risk factors and mechanisms driving disease development [4]. In this context, the immune system closely interacts with the nervous system and plays a pivotal role in ALS and AD, where it appears to amplify disease progression. Both peripheral immune cells (e.g., T cells and monocytes) and CNS-resident microglia are involved, with emerging evidence also linking the immune system to the gut microbiome [5]. Furthermore, biological sex has been proposed to play a role in modulating immune cell function in AD and ALS [2]. However, most immune cell profiling studies have focused on prevalent cases [5], making it challenging to disentangle cause–effect relationships. While therapies targeting specific immune cell subtypes hold promise, the analysis of prevalent cases limits insights into the preclinical stages, where interventions would likely be most effective. In this context, our group recently investigated possible alterations of a subset of effector memory CD8+ T cells, specifically CD8+ effector memory T cells re-expressing CD45RA (TEMRA) cells, in healthy individuals at high genetic risk of AD [6], based on prior reports suggesting differential levels of these cells in pre-clinical AD cases [7]. Although these analyses did not confirm the role of CD8+ TEMRA cells in the early pre-clinical phase of AD, the fact that changes in immune cell composition associated with AD or ALS are likely to precede the clinical onset of these diseases by years prompted us to investigate early immune cell alterations more comprehensively, using the same cohort as in our previous study [6]. To this end, we investigated whether healthy individuals at high genetic risk of these diseases show alterations in relevant immune cell subtypes. Specifically, we calculated polygenic risk scores (PGSs) for ALS and AD and examined their associations with 92 immune cell subtypes in ~450 individuals from the Berlin Aging Study II (BASE-II). Given the potential modifying role of biological sex in immune cell function in AD and ALS [2], we also investigated sex-specific effects for our top results in men and women.

## 2. Methods

### 2.1. Study Participants

A total of 448 European individuals recruited as part of BASE-II were included in this study (described in ref. [6]). BASE-II is a multi-institutional longitudinal cohort study from the larger Berlin area in Germany that investigates determinants and mechanisms of aging (described in detail elsewhere [8]). The effective dataset investigated here consists of individuals from two age groups, comprising 309 older adults (60% females, median age: 69 years, age range: 60–82 years) and 139 younger adults (59% females, median age: 29, age range: 23–35, Appendix A). All individuals were immunologically healthy (no fever or immune system-related diseases at the time of sampling), i.e., participants with immune system-related diseases or treatment at baseline were excluded, comprising autoimmune diseases (rheumatoid arthritis, psoriatic arthritis, systemic lupus erythematosus, type 1 diabetes, multiple sclerosis, Crohn’s disease, ulcerative colitis, psoriasis, celiac disease, or lymphocytic colitis), human immunodeficiency virus, acute fever (defined as current or within the previous six weeks), and receiving immunomodulatory systemic therapy. In addition, participants with CRP values >10 mg/L, as measured in plasma drawn at baseline, were excluded. Furthermore, participants with mild cognitive impairment and/or dementia at baseline were excluded based on Mini-Mental State Examination testing (MMSE), as described in ref. [6]. Please note that an unauthorized version of the German MMSE was utilized by the study team without permission, and has been rectified with PAR (https://www.parinc.com (accessed on 20 December 2024)). The MMSE is a copyrighted assessment tool and may not be used, reproduced, or distributed, in full or in part, in any form, language, or format without prior written permission from PAR. Moreover, participants with self-reported ALS and Parkinson’s disease at baseline were excluded.

All participants provided written informed consent and the study was conducted in accordance with the Declaration of Helsinki and approved by the Ethics Committee of the Charité—Universitätsmedizin Berlin—approval number EA2/029/09.

### 2.2. Generation and Processing of Immune Cell and Genome-Wide SNP Data

Peripheral blood mononuclear cells were isolated and analyzed by multicolor flow cytometry, as described previously [6,9]. Briefly, two different immune cell antibody panels were used to distinguish between various immune cell subtypes, i.e., subsets of T cells, B cells, natural killer cells, or monocytes, with a particular emphasis on T cell subtypes (Appendix A [10]). Specifically, we included CD45RA and CCR7 markers for analysis of T cell differentiation based on the well-established model of CD45RA and CCR7 expression [11]. Additionally, we included CD27 and CD28 markers, as more refined models have highlighted their utility in better defining T cell differentiation states [12,13]. Thus, to distinguish naïve T cells from late-differentiated T cells, both of which express CD45RA, we relied on the differential expression of CD27 and CD28. To further validate our findings, we analyzed CD57 expression. The anti-CD57 antibody identifies senescent CD57+ cells, in contrast to naïve T cells, which usually lack CD57 expression [13]. We also examined the rare population of T memory stem cells, characterized by their stem cell-like ability to self-renew, and their capacity to regenerate the entire range of memory and effector T cell subsets. It has been proposed that CD95 and IL-2Rβ expression on otherwise phenotypically naïve T cells can identify T memory stem cells [14]. To identify potentially exhausted T cells, we analyzed the expression of the inhibitory receptor ‘programmed cell death protein 1′ (PD-1) [15,16]. PD-1 has been shown to be present on T cells infiltrating the brain [17]. A 3-laser BD LSRII (BD biosciences) flow cytometer and DIVA6 software (https://www.bdbiosciences.com/en-de/products/software/instrument-software/bd-facsdiva-software, accessed on 20 December 2024) were utilized for the acquisition of the data, analyzed thereafter using FlowJo version 7.5, as described previously [6,9]. QCed immune cells were quantified as proportions of the major immune cell types and distributions were transformed where appropriate (based on visual inspections) using one of the following approaches: log10, root, log(100 − x), or square [18] (Appendix A [10]). As a final step, the data were z-transformed.

The genome-wide SNP data were generated and processed as described before [6]. In short, the Affymetrix Array 6.0 was used for genotyping, followed by the imputation of unmeasured genotypes using the Haplotype Reference Consortium reference panel, and was finalized using standard QC procedures [6]. After filtering for a minor allele frequency (MAF) threshold of <0.01 and an imputation quality threshold of <0.3, we obtained a final set of 7,512,709 SNPs. As the imputation of the established AD *APOE* risk SNP e4 (rs429358) was suboptimal in this dataset (imputation r2 = 0.64), this SNP was directly genotyped using “Taqman” technology (Thermo Fisher Scientific, Inc., Waltham, MA, USA) [6].

The calculation of the PGSs for ALS was based on 11 index SNPs that showed genome-wide statistical significance in European individuals from a recent large genome-wide association study (GWAS) on ALS [19] (Appendix A, Appendix A). By including these 11 SNPs, we relaxed the minor allele frequency threshold for 1 SNP (rs80265967, MAF = 0.00039) to maximize the number of SNPs included in the PGSs. Using sensitivity analysis, we re-calculated an additional PGS and subsequent association statistics excluding this SNP (Appendix A). PGS calculations for AD were based on 70 genome-wide significant SNPs (α = 5.0 × 10^−8^) from a recent GWAS on AD [20]. Of note, the beta estimate and *p* value of the *APOE* e4 SNP were taken from an earlier AD GWAS [21] by the same group, because it was excluded from the summary statistics in ref. [20] (Appendix A). Due to the missing data of the *APOE* e4 genotype in some BASE-II participants, some individuals had to be excluded from the analyses (*n* = 18), resulting in a maximum of 430 individuals included in the AD analyses (Appendix A). The PGS was z-transformed prior to statistical analyses.

### 2.3. Statistical Analyses

Linear regression analyses using the z-transformed PGSs as exposure and the z-transformed immune cell proportions as outcome were performed, while adjusting for sex, age group, and the first four genetic principal components to account for potential population substructures. The analyses were conducted using the ‘lm’ function in R. The main analyses were performed on the full sample and the subgroup of older BASE-II participants. Additionally, exploratory analyses stratifying for sex were performed on all nominally significant results (α = 0.05). Sensitivity analyses included an adjustment for cytomegalovirus (CMV) status and the utilization of a modified PGS for ALS. The threshold for the false discovery rate (FDR) was set to 5%.

## 3. Results

The linear regression analyses of PGSs on 92 immune cell subtypes in the full sample and the subgroup of older individuals did not yield significant signals at an FDR of 5%, neither for ALS nor for AD (Appendix A). Despite the lack of study-wide significant findings, we observed several nominally significant (α = 0.05)—and hence potentially relevant—signals:

For ALS, five cell-type proportions passed the nominal threshold in analyses of the full BASE-II sample (Figure 1). This included the T cell subsets CD57+ CD8+ early-memory T cells in total CD8+ T cells (*p* = 0.006), PD-1+ CD8+ T cells in total CD8+ T cells (*p* = 0.024), and CD57+ CD4+ early-memory T cells in total CD4+ T cells (*p* = 0.038), as well as non-classical monocytes in total monocytes (*p* = 0.015) and myeloid dendritic cells in total monocytes (*p* = 0.047), with variances explained (∆R2) ranging from 2.1% to 1.1% (Table 1). Notably, CD57+ CD8+ early-memory T cells exhibited consistent effect estimates across all analyses, and importantly, showed the strongest effect estimates in men, with the PGS explaining 7% of the variance in immune cell distribution in this subset (*p* = 0.001). The same cell type also reached nominal significance when analyses were restricted to the older age group (∆R2 = 2.4%, *p* = 0.012). In addition, PD-1+ CD8+ T cells, also exhibiting consistent effect estimates across strata, showed a nominally significant association not only in the full sample, but also in the subset of men (∆R2 = 2.6%, *p* = 0.041; Table 1). However, despite the more pronounced effect estimates in men, their confidence intervals were largely overlapping for both cell types (Figure 1). Notably, there were moderate to strong correlations among some cell types nominally significantly associated with ALS risk. Specifically, CD57+ CD8+ early-memory T cells were correlated with CD57+ CD4+ early-memory T cells (r = 0.62) and CD8+ PD-1+ cells (r = 0.46). Additionally, non-classical monocytes exhibited moderate correlations with myeloid dendritic cells (r = 0.33). The other cell types nominally significantly associated with ALS risk showed no or only weak correlations (r < 0.3; Figure 1).

For AD, we did not observe any trends of association (α = 0.05) in the full BASE-II dataset. However, upon restricting our investigation to the older subsample, i.e., that closer to a potential clinical onset of AD, we observed trends toward an association of higher genetic AD risk with subsets of CD8+ T cell and natural killer cell proportions (Figure 1). Specifically, a higher PGS was associated with increased naïve CD57+ CD8+ T cells in total CD8+ T cells (*p* = 0.015), increased mature NKG2A+ natural killer cells in total leukocytes (*p* = 0.024), and decreased CD8+ late effector memory T cells in total CD8+ cells (*p* = 0.038), with ∆R2 ranging from 2.4% to 1.7%. The first two associations were more pronounced in the male subgroup, while the last was stronger in women, but confidence intervals were largely overlapping (Table 1). In contrast to ALS, none of the cell types associated with AD risk showed considerable correlations (r < 0.2; Figure 1).

Lastly, none of the results changed appreciably in the sensitivity analyses when adjusting for CMV or using the alternative PGSs for ALS (Appendix A).

## 4. Discussion

In this study, we performed a comprehensive investigation of whether alterations in peripheral blood immune cell composition are evident in healthy participants at high genetic risk of AD or ALS. While none of the associations reached study-wide significance in ~450 BASE-II participants after FDR control, we observed several nominally significant associations between genetic ALS and AD risk, and various immune cell subsets. These include subsets of CD4+, CD8+ T cells, monocytes, and myeloid dendritic cells for ALS, as well as subsets of CD8+ T cells and natural killer cells for AD. To our knowledge, this is the first study to analyze such a broad spectrum of immune cell subtypes in healthy individuals at high risk of ALS or AD. The vast majority of previous studies investigating immune cells in ALS and AD analyzed prevalent cases [5].

Notably, the nominally significant results showed consistent associations overall across the analyzed strata. The most compelling results for ALS suggest a decrease in CD57+ CD8+ early-memory T cells and in PD-1+ CD8+ T cells. Although both cell types exhibited stronger effects in men than in women in our study, the largely overlapping confidence intervals of the effect estimates prevent definitive conclusions. Thus, at this stage, it remains unclear whether these potential sex differences are due to chance, or if they indicate male-specific pathophysiological mechanisms. The latter possibility would be particularly intriguing given the increased ALS risk observed in men, and the known influence of biological sex on immune cell function [2]. Interestingly, a recent landmark study investigated both effector and memory CD8+ T cells in human monogenic ALS patients with *SETX* mutations, as well as PD-1+ CD8+ T cells in mice [22]. The authors found a general decrease in CD8+ effector and memory T cells, which is in agreement with our observations. While they did not show data for PD-1+ CD8+ T cells in humans, they described differential proportions of these cells in the spinal cord, brain, and blood of a murine ALS model. However, effect directions differed in mice vs. our human data, potentially reflecting the limited comparability between mouse models and humans [22]. The observed relative decrease in CD57+ CD8+ early-memory T cells in this study is challenging to interpret, as this population itself represents a heterogeneous group of cells. This reduction might indicate a decline in the pool of functional, mature memory T cells, potentially compromising the ability to mount an adequate immune response [23]. The function of PD-1+ CD8+ T cells is also context dependent, with expression of the inhibitory receptor PD-1 serving as a modulator of T cell activity and tolerance [24], and being expressed on T cells infiltrating the brain [17]. In this context, it is relevant that some of the T cell types associated with ALS risk appear to be partly co-regulated or interdependent in our dataset. In particular, CD8+ early-memory cells and CD8+ PD-1+ T cells showed strong correlations, as did CD8+ and CD4+ early-memory T cells. At this stage, it remains uncertain whether one may play a more functionally relevant role than others in the potential link to ALS risk.

Apart from associations with T cell distributions, myeloid dendritic cells and non-classical monocytes also nominally significantly decreased in individuals at higher genetic risk of ALS. The latter finding supports previous reports, where the functional role of peripheral monocytes early in the disease process was postulated [5,25]. In summary, if validated, these findings support previous observations that in addition to the involvement of the innate immune system (monocytes), T cell-mediated immune responses also play a role in the pathophysiology of ALS [5].

In individuals at high genetic risk of AD, the noteworthy immune cell alterations highlighted by our analyses relate to increased naïve CD57+ CD8+ T, mature NKG2A+ natural killer, and CD8+ late effector memory T cells. Consistent effect directions were observed for the main and all subgroup analyses. Similarly to ALS, the wide confidence intervals of the sex-stratified analyses preclude any definitive conclusions regarding underlying sex-specific biological mechanisms. The population of naïve CD57+ CD8+ T cells remains largely unexplored in AD and other conditions, making any assumptions for the relevance of the observed association complicated. The presence of CD57 on naïve CD8+ T cells is unusual because CD57 is typically associated with terminal differentiation, reduced proliferation, and a senescent-like phenotype, which is not characteristic of naïve T cells [26]. It remains speculative whether these cells represent atypical peripheral T cell phenotypes potentially impacted by neuroinflammatory processes or whether they reflect a transitional state between naïve and differentiated T cell subsets. However, a number of studies reported on the relevance of other subtypes of CD8+ T cells in AD, which are altered in the periphery and infiltrate into cerebrospinal fluid and into the brain [5,27]. Our nominally significant finding for NKG2A+ natural killer cells may imply their immune regulatory mechanisms in AD risk; NKG2A, if expressed on natural killer cells, binds to HLA class I molecules on healthy, intact cells and prevents cytotoxic attacks. However, when the target cells become infected and lose their surface HLA-I molecules, the inhibitory effect is lost, and the natural killer cells attack the defective cells [28]. In this context, while the role of natural killer cells in the pathogenesis of AD has been reported repeatedly [27,29], the role of the different subsets in AD and the corresponding mechanisms remain largely elusive at this point.

Overall, it is noteworthy that we found differences in immune cells for individuals at risk of AD only in the smaller older age group and not in the overall group, a pattern which we did not observe for ALS risk. This difference may be attributable to the approximately ten-year-later age of onset of AD compared to ALS. It can only be speculated that peripheral immune cell alterations may be more pronounced in AD closer to the onset of clinical symptoms compared to several years earlier.

The primary strengths of this study are an extensive and thorough clinical examination of all BASE-II participants, in-depth immune cell phenotyping, and a comparatively large sample size.

Potential limitations of our work include limited statistical power to detect minor effects, the missing consideration of lifestyle and environmental ND risk factors, and the lack of data in non-European individuals, as discussed in previous work by our group [10]. Furthermore, it is important to note that the PGS for AD is primarily driven by APOE ε4, the most significant genetic risk factor for AD, while non-*APOE* risk SNPs contribute only modestly to the overall risk score [30]. A separate analysis of homozygous ε4/ε4 carriers versus non-carriers could offer valuable insights into the role of *APOE*, particularly in light of recent research on this topic [31]. However, the limited number of *APOE* ε4 homozygous carriers (*n* = 8) precluded sufficiently powered analyses in this dataset. In our previous study [10], we examined alterations in immune cell types in healthy individuals with a high genetic risk of Parkinson’s disease. Together with that study, our findings suggest that major changes in immune cell composition may not manifest in healthy individuals with an increased genetic risk of the investigated NDs. Given the well-established role of the immune system response in ND pathogenesis [32], this suggests that substantial immune cell alterations become manifest at later stages of disease development. Notwithstanding, our work highlights interesting trends of alterations in several innate and adaptive immune cell subtypes, which warrant further investigation in future research.

## 5. Conclusions

We found some intriguing nominally significant results particularly for ALS, which need to be replicated in independent studies. Overall, our findings suggests that major immune cell changes predominately manifest at later disease stages in ALS and AD.

## Figures and Tables

**Figure 1 cells-14-00250-f001:**
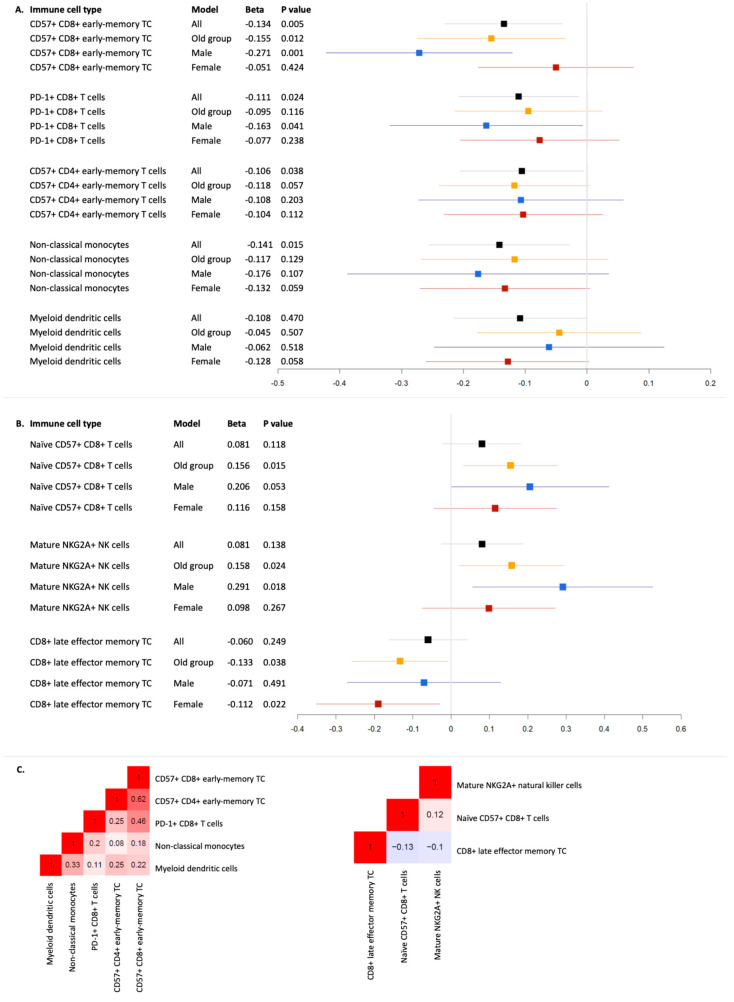
Nominally significant linear regression results of neurodegenerative disease polygenic risk scores with immune cell distributions in the blood of healthy BASE-II participants. Legend. This forest plot displays the beta values and 95% confidence intervals of nominal significant association results linking amyotrophic laterals sclerosis (**A**) and Alzheimer’s disease (**B**) and levels of immune cell subtypes in blood as well as Spearman’s correlation coefficients between the corresponding nominally significant cell types per disease (**C**). TC = T cells and NK = natural killer cells.

**Table 1 cells-14-00250-t001:** Immune cells showing nominally significant associations with genetic risk profiles of amyotrophic lateral sclerosis and Alzheimer’s disease.

	Immune Cell Type	Model	Beta	∆R2	*p*
**ALS**	**CD57+ CD8+ early-memory T cells**	**All**	−0.135	0.018	**0.006**
	**CD57+ CD8+ early-memory T cells**	**Old group**	−0.155	0.024	**0.012**
	CD57+ CD8+ early-memory T cells	Female	−0.051	0.003	0.424
	**CD57+ CD8+ early-memory T cells**	**Male**	−0.271	0.072	**0.001**
	**PD-1+ CD8+ T cells**	**All**	−0.111	0.012	**0.024**
	PD-1+ CD8+ T cells	Old group	−0.095	0.009	0.116
	PD-1+ CD8+ T cells	Female	−0.077	0.006	0.238
	**PD-1+ CD8+ T cells**	**Male**	−0.163	0.026	**0.041**
	**Non-classical monocytes**	**All**	−0.141	0.021	**0.015**
	Non-classical monocytes	Old group	−0.117	0.015	0.129
	Non-classical monocytes	Female	−0.133	0.018	0.059
	Non-classical monocytes	Male	−0.176	0.031	0.107
	**CD57+ CD4+ early-memory T cells**	**All**	−0.106	0.011	**0.038**
	CD57+ CD4+ early-memory T cells	Old group	−0.118	0.014	0.057
	CD57+ CD4+ early-memory T cells	Female	−0.104	0.011	0.112
	CD57+ CD4+ early-memory T cells	Male	−0.108	0.011	0.203
	**Myeloid dendritic cells**	**All**	−0.108	0.012	**0.047**
	Myeloid dendritic cells	Old group	−0.045	0.002	0.507
	Myeloid dendritic cells	Female	−0.128	0.017	0.058
	Myeloid dendritic cells	Male	−0.062	0.004	0.518
**AD**	Naïve CD57+ CD8+ T cells	All	0.081	0.007	0.118
	**Naïve CD57+ CD8+ T cells**	**Old group**	0.156	0.024	**0.015**
	Naïve CD57+ CD8+ T cells	Female	0.116	0.013	0.158
	Naïve CD57+ CD8+ T cells	Male	0.206	0.039	0.053
	Mature NKG2A+ natural killer cells	All	0.081	0.007	0.138
	**Mature NKG2A+ natural killer cells**	**Old group**	0.158	0.024	**0.024**
	Mature NKG2A+ natural killer cells	Female	0.098	0.009	0.267
	**Mature NKG2A+ natural killer cells**	**Male**	0.291	0.079	**0.018**
	CD8+ late effector memory T cells	All	−0.060	0.004	0.249
	**CD8+ late effector memory T cells**	**Old group**	−0.133	0.017	**0.038**
	**CD8+ late effector memory T cells**	**Female**	−0.112	0.013	**0.022**
	CD8+ late effector memory T cells	Male	−0.071	0.005	0.491

Legend. This table visualizes the results of all immune cell analyses that show nominally significant (α = 0.05) associations in at least one analysis (i.e., analyzing all or restricting to the older group). The linear regression analyses were performed across 448 healthy individuals from the Berlin Aging Study II (BASE-II) using polygenic scores (PGSs) for amyotrophic laterals sclerosis (ALS) and Alzheimer’s disease (AD), with exposure and immune cell proportions as outcome. Bold indicates nominally significant association results. Beta = effect estimate of regression analyses of z-transformed immune cell distributions on z-transformed polygenic risk scores. ∆R2 = variance of immune cell distribution explained by the polygenic risk score.

## Data Availability

All summary statistics have been made available in the Appendix A of this manuscript. Raw and source data are available upon reasonable request. Interested researchers may contact the scientific BASE-II coordinator, Ludmila Müller, lmueller@mpib-berlin.mpg.de. Additional information is available on the BASE-II website: https://www.base2.mpg.de/7549/data-documentation (accessed on 20 December 2024).

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
