# Peer review of "Peripheral Immune Profiles in Individuals at Genetic Risk of Amyotrophic Lateral Sclerosis and Alzheimer’s Disease"

_cells, 2025, doi:10.3390/cells14040250_

Round 1
Reviewer 1 Report
Comments and Suggestions for Authors
The manuscript by Deeke et al. reports new interesting details about the immune cell types in AD and ALS. Overall the study is well designed and presented and of interest to the community.
I only would suggest to the authors to go in a bit more detail about the differences between male versus female patients in AD versus ALS first in the introduction, and then also in the discussion regarding the findings for the different immune cell types and what kind of significance this could have regarding the pathologies.
Reviewer 2 Report
Comments and Suggestions for Authors
The study is interesting and expands knowledge about immunological processes in neurodegenerative diseases.
I have a few comments on this manuscript, which are presented below:
1. The authors should better justify the choice of neurodegenerative diseases - why did they decide to study people with a predisposition to ALS/AD?
2. Methods: please justify the basis for excluding people with a diagnosis of neurodegenerative disease - were any tests performed, and if so, which ones?
3. Methods: please indicate the basis for excluding people with diseases related to the immune system - which diseases were excluded and on what basis
4. Methods: no information regarding the consent of the ethics committee and the consent of participants to participate in the study
5. The association of the APOE e4 genotype with the risk of AD is relatively weak and dependent on the homo- (stronger association)/heterozygous (weaker association) status - this should be included in the analyses; perhaps only homozygotes should be analyzed
6. Genetic determinants of ALS are also relatively weak and it is difficult to classify healthy individuals into a high-risk group based on them; so far, one of the genes most strongly associated with the risk of ALS is SOD1 - was it included in the analyses?
7. In the discussion, the authors should not only describe their observations and relate them to previously published research results, but also refer to the observed relationships - what significance may the noted changes in immunological parameters have? why were the changes slightly more pronounced in the older group and among men or women. Why may there be differences between individuals with a predisposition to AD/ALS? etc.
Reviewer 3 Report
Comments and Suggestions for Authors
Summary and overall impression
In this study by Deecke et al., the authors analysed peripheral immune cell phenotypes in 448 healthy European individuals with genetic predisposition to ALS and AD. ALS risk was significantly associated with decreased CD57+ CD8+ early memory T cells, PD-1+ CD8+ T cells, CD57+ CD4+ early memory T cells, non-classical monocytes and myeloid dendritic cells. AD risk was significantly associated with increased naïve CD57+ CD8+ T cells and mature NKG2A+ NK cells. The study is interesting and useful, as it is among the first of its kind to analyse immune cell subtypes in a large sample size of healthy individuals with a high genetic risk for ALS and AD. The manuscript would benefit from a more detailed explanation for the choice of markers used in immune cell phenotyping.
Specific comments
1. Abstract, line 52 and 55: The authors mention CD8+ TEMRA cells, but there is no explanation of the abbreviation and its context in relation to the current study.
2. Immune cell phenotyping: It would be good if the authors can briefly explain their panel design in the manuscript, especially the inclusion of less common markers such as CD57, CD27, CD95, PD-1 etc. What were the authors’ rationale for including these markers when distinguishing cell populations?
3. Supplementary Tables 1 and 2: I note that the authors had included CD33 in Panel 2 of Supp. Table 2, but I don’t see CD33 listed as one of the markers in their list of 92 immune cell subtypes.
4. Figure 1: Labelling of the immune cell types can be improved for the benefit of all readers (especially non-immunologists!), instead of using abbreviations such as M_N, ME, LEM.
5. Figure 1 legend, line 141: should be “levels of immune…”
6. Figure 1C: I note that the authors include a chart for the correlation of the nominally significant cell types in ALS and AD, but there is no further description or explanation of it in the text. Is Figure 1C necessary and does it help in further understanding the data?
7. Discussion: I would have liked to see the authors elaborate more on the possible implications of the immune cell subtypes that were significantly associated with ALS and AD risk. For example, what is the role of PD-1+ CD8+ T cells in disease and what can a decrease in these cells in healthy individuals at risk of disease possibly mean? Again, this can tie in with the authors’ rationale for including markers such as CD57 and PD-1 in their panel.
Round 2
Reviewer 3 Report
Comments and Suggestions for Authors
The authors have made good efforts to address all of the reviewers’ comments and improve their manuscript.
Minor comments:
1. Line 123 “The anti-CD57 antibody identifies senescent CD57⁺ cells…”: the phrasing of this sentence is a little clumsy. Consider re-phrasing as “Senescent cells are positive for CD57, in contrast to naïve T cells…”.
2. Line 128 “stem cells cells”: typo error.